

# Treating coral bleaching as weather: a framework to validate and optimize prediction skill

Thomas M. DeCarlo

[1] Hawaii Pacific University, Honolulu, HI, United States of America
[2] Red Sea Research Center, Division of Biological and Environmental Science and Engineering, King Abdullah University of Science and Technology, Thuwal, Saudi Arabia

## ABSTRACT

Few coral reefs remain unscathed by mass bleaching over the past several decades, and much of the coral reef science conducted today relates in some way to the causes, consequences, or recovery pathways of bleaching events. Most studies portray a simple cause and effect relationship between anomalously high summer temperatures and bleaching, which is understandable given that bleaching rarely occurs outside these unusually warm times. However, the statistical skill with which temperature captures bleaching is hampered by many "false alarms", times when temperatures reached nominal bleaching levels, but bleaching did not occur. While these false alarms are often not included in global bleaching assessments, they offer valuable opportunities to improve predictive skill, and therefore understanding, of coral bleaching events. Here, I show how a statistical framework adopted from weather forecasting can optimize bleaching predictions and validate which environmental factors play a role in bleaching susceptibility. Removing the 1 °C above the maximum monthly mean cutoff in the typical degree heating weeks (DHW) definition, adjusting the DHW window from 12 to 9 weeks, using regional-specific DHW thresholds, and including an El Niño threshold already improves the model skill by 45%. Most importantly, this framework enables hypothesis testing of other factors or metrics that may improve our ability to forecast coral bleaching events.

## INTRODUCTION

Coral bleaching, the loss of the symbiotic zooxanthellae algae that live within coral tissues, is recognized as one of the primary threats facing reef-building corals today. Although bleaching is occasionally observed in isolated or scattered coral colonies, "mass bleaching events" occur when multiple species across entire reefs bleach at once. The link between mass coral bleaching and unusually warm water temperature was established in the wake of the widespread 1982-83 El Niño, which caused severe bleaching across the Pacific Ocean (*Glynn, 1983*; *Glynn, 1993*; *Coffroth, Lasker & Oliver, 1990*). Since then, bleaching events have become more frequent (*Hughes et al., 2018*), and have continued to coincide with anomalously high temperatures (e.g., *Bruno et al., 2001*; *DeCarlo et al., 2017*; *Donner et al.,*

Corresponding author
Thomas M. DeCarlo,
tdecarlo@hpu.edu

*2017*; *Hughes et al., 2017*; *Barkley et al., 2018*; *Sully et al., 2019*). Yet, some high-temperature events have not caused bleaching (*Thompson & Van Woesik, 2009*; *Pratchett et al., 2013*; *Gintert et al., 2018*; *DeCarlo & Harrison, 2019*). The absence of bleaching is often not included in large-scale analyses of coral bleaching (but see e.g., *Sully et al., 2019*), which precludes a statistical validation of the skill of temperature in predicting bleaching events. Rather, our ability to predict bleaching, and to understand the causes of these events, depends on correctly capturing *both* the presence and absence of bleaching.

One way to critically assess our understanding of the causes of coral bleaching is to view these events within a weather-forecasting framework (*Van Hooidonk & Huber, 2009*). Meteorologists have a strong understanding of the factors that cause precipitation, and they are able to predict when and where there will be rain or snow based on wind, temperature, and pressure patterns. Intuitively, we want to know both when rain will fall, and when it will not. Forecasting rain when it is actually sunny is deemed just as bad as forecasting clear skies when it actually rains. We should view coral bleaching in the same way. Fortunately, the statistical tools to do so are already in place because weather forecasters have been validating and optimizing their predictions for decades. Here, I apply weather forecasting statistics to a global coral bleaching database to (1) evaluate the skill of bleaching predictions, and (2) explore various approaches to improve prediction skill. Additionally, since the choice of sea surface temperature (SST) product affects bleaching predictions (*DeCarlo & Harrison, 2019*), as a first step I evaluate the agreement between various SST products in terms of representing global mean anomalies.

## METHODS

To assess the agreement between the various SST products that could be used here to predict coral bleaching, I calculated global-mean annual SST anomalies using six datasets:

Coral Reef Watch's CoralTemp (CRW) (*Liu et al., 2014*), Optimum Interpolation SST (OI-SSTv2) (*Reynolds et al., 2007*; *Banzon et al., 2016*), the Canadian Meteorological Center (CMC) (*Canada Meteorological Center, 2012*), the Extended Reconstructed SST version 5 (ERSSTv5) (*Huang et al., 2017*), the Met Office Hadley Centre SST version 4 (HadSST4) (*Kennedy et al., 2019*), and the Japan Meteorological Agency (JMA) Centennial In Situ Observation-Based Estimates of SST (COBE2) (*Hirahara, Ishii & Fukuda, 2014*). Comparing global-mean annual SST anomalies is a coarse evaluation since it does not account for potentially important differences on relatively small spatial and temporal scales. Nevertheless, this type of analysis is common for evaluations of SST products (*Huang et al., 2018*), and it can identify the broad-scale differences among products. I performed a principal component analysis (PCA) to evaluate how SST products cluster together. Additionally, I calculated pair-wise annual differences between the products to identify any consistent discrepancies between products.

The key to applying weather forecasting statistics to coral bleaching is a database complete with both presence and absence of bleaching, which only recently became available (*Hughes et al., 2018*). Other databases, such as the bleaching reports in ReefBase, are unsuitable because they are dominated by reports of bleaching and include relatively few

**Table 1   Schematic contingency table of observed and predicted events.**

| | | Observed | |
|---|---|---|---|
| | | yes | no |
| Predicted | yes | Hits ("$H$") | False Alarms ("$FA$") |
| | no | Misses ("$M$") | Correct Negatives ("$CN$") |

observations of the absence of bleaching (e.g., see *Logan et al., 2012*). The complete database of *Hughes et al. (2018)*, however, enables the application of binary (presence/absence) event detection metrics that are commonly used to evaluate weather forecasts. In addition to bleaching absences, this database includes "moderate" (1–30% bleaching) and "severe" (greater than 30% bleaching) events, but for most analyses here the moderate and severe events are combined to reduce the data to bleaching presence/absence (but see exception below when only severe events are considered). All of the binary metrics are based on a simple contingency table of predictions versus observations (Table 1). No single metric can capture all the information in the dataset, and thus a variety of metrics are used to evaluate predictive skill (Table 2). Together, these metrics can be used to validate whether a predictor performs better than random chance, and to optimize the threshold of a continuous predictor used to forecast events. Metrics focused on one aspect of the prediction skill (Accuracy, Probability of Detection, Bias, Probability of False Detection, and False Alarm Ratio) generally change monotonically with the DHW threshold. Thus, while these metrics hold key information, they do not clearly provide an optimal threshold. Conversely, the Equitable Threat Score (ETS) accounts for multiple aspects of the forecast, rewarding correct predictions and penalizing incorrect ones (*Hamill, 1999*). ETS is the most commonly used threshold optimizer in weather forecasts (*ECMWF, 2018*), and it is adopted here as a guide for coral bleaching predictions. It is worth noting that ETS and the related Pierce's Skill Score have previously been applied to evaluate coral bleaching (*Van Hooidonk & Huber, 2009*; *Mollica et al., 2019*), but not yet on a dataset like *Hughes et al. (2018)* that is complete with both presence and absence. Additionally, there are other skill scores in use (Table 2), which are insightful under some circumstances, but are generally considered less robust and are used less frequently by weather forecasters.

   The SST product OI-SSTv2 was identified as the most appropriate for this analysis due to its relatively high resolution (∼25-km and daily), temporal coverage from 1982 to present, and agreement with most other SST datasets (see Results and Discussion). Further supporting the use of OI-SSTv2 here is that it is designed to minimize temporal bias by using a single type of sensor, the advanced very high resolution radiometer (AVHRR) (*Banzon et al., 2016*). Indeed, in a comparison of satellite SST to In Situ reef-water temperature measured by loggers on the Great Barrier Reef, OI-SSTv2 was more consistent over time than other SST products, especially in terms of representing the highest summer temperatures (*DeCarlo & Harrison, 2019*).

   Here, I used OI-SSTv2 as a predictor of coral bleaching events at 100 globally-distributed reef sites between 1982 and 2016 (using the database of *Hughes et al., 2018*). I calculated degree heating weeks (DHW) from OI-SSTv2, defining the climatological maximum

**Table 2** Metrics to assess the binary (presence/absence) detection of events. The various metrics, formulas, meanings, and ranges are displayed.

| Test | Formula | Meaning | Range |
|---|---|---|---|
| Accuracy | $(H+CN)/n$ | Fraction of predictions correct | 0 to 1, 1 = perfect, 0.5 = no skill |
| Bias | $(H+FA)/(H+M)$ | Predicted "yes" relative to observed "yes" | 0 to infinity, 1 = perfect |
| Probability of Detection (PD) | $H/(H+M)$ | Fraction observed "yes" were predicted | 0 to 1, 1 = perfect, 0.5 = no skill |
| Probability of False Detection (PFD) | $FA/(CN+FA)$ | Fraction of observed "no" were predicted "yes" | 0 to 1, 0 = perfect |
| False Alarm Ratio | $FA/(H+FA)$ | Fraction of predicted "yes" were not observed | 0 to 1, 0 = perfect |
| Equitable Threat Score (ETS) | $H/(H+FA+M-H_{random})$ | Correspondence of predicted events to observed events, but accounting for hits due to random chance | $-1/3$ to 1, 1 = perfect, 0 = no skill |
| Threat Score (TS) | $H/(H+FA+M)$ | Correspondence of predicted events to observed events | 0 to 1, 1 = perfect, 0 = no skill |
| Odds Ratio Skill Score (ORSS) | $(H*CN-M*FA)/(H*CN+M*FA)$ | The improvement of predictions above random chance | $-1$ to 1, 1 = perfect, 0 = no skill |
| Pierce's Skill Score | $H/(H+M)-FA/(FA+CN)$ | Separation of events from non-events | $-1$ to 1, 1 = perfect, 0 = no skill |

**Notes.**
$H_{random} = (H+FA)*(H+M)/n$, where $n$ is the total number of observations (3,500 in this case).

monthly mean (MMM) based on 1982-2012. DHW incorporates both the magnitude and duration of SST anomalies above the MMM, and is commonly used as a predictor of coral bleaching (*Liu, Strong & Skirving, 2003*; *Liu et al., 2006*; *Hughes et al., 2017*). Since many of the reef areas in the database are larger than 25 km, I used the maximum annual DHW among the grid-box closest to the database coordinates and its neighbors. The database as published in *Hughes et al. (2018)* required fixing coordinates for 13 out of 100 sites (Table S1). Additionally, bleaching in the central and northern Great Barrier Reef was incorrectly labeled in the original database as moderate in 1983, when the literature on this event indicates that it was instead a severe event in 1982 (*Fisk & Done, 1985*; *Harriott, 1985*; *Oliver, 1985*). Likewise, 1987 bleaching in the Galápagos Islands was reported as severe, but the literature clearly shows that this was a moderate event (*Podestá & Glynn, 1997*; *Glynn et al., 2001*; *Glynn et al., 2017*). Finally, no bleaching occurred in the Al Lith region of the Red Sea during 2010 (event removed because no literature supports this event). The adjustments I made to the bleaching histories from the database as published in *Hughes et al. (2018)* improved the statistical skill, albeit only slightly since my changes affected relatively few entries in the database (Tables S2–S3). Even though the database was presented as being temporally complete (i.e., the authors implied that the full history of bleaching during 1980–2016 was known at each of the 100 sites), it is possible that other inaccuracies exist in the database, including either mislabeling as noted above or bleaching events that went unnoticed. Nevertheless, except for the noted corrections, I took the database at face value since no alternative global databases are suitable for this study. Additionally, I explored whether patterns in the database are indicative of unnoticed

bleaching events in the early part of the database by evaluating whether (1) false alarms decrease and (2) ETS increases over time.

The common definition of DHW includes a 1 °C above the MMM cutoff for accumulating heat stress, and DHW are calculated over the preceding 12 weeks. In initial analyses, I tested whether removing the 1 °C cutoff (i.e., DHW begin accumulating as soon as SST exceeds the MMM), and using time windows shorter than 12 weeks, improved the results. I also varied the DHW threshold used to predict bleaching between 1 and 15 °C-weeks to find the optimal value. In subsequent analyses, I evaluated whether prediction skill increased when using regional-specific DHW thresholds, and including an additional El Niño threshold (based on the Niño3.4 SST anomaly definition of El Niño). Additionally, I evaluated if maximum Hotspots (SST anomalies above the MMM) or the Marine Heatwave index (*Hobday et al., 2018*) performed better than DHW since some analyses of coral bleaching events have found these shorter-term metrics to be most effective (*Berkelmans et al., 2004*; *Genevier et al., 2019*). Finally, I tested whether there is evidence for an increase in the bleaching threshold over time.

## RESULTS AND DISCUSSION

### Choice of SST product

A variety of SST products are available that include a range of spatial resolution ($\sim$5 km to over 200 km), temporal resolution (daily to monthly), temporal coverage (several decades to beyond a century), and methods for bias correction. These products can broadly be classified into two groups: coarse-resolution centennial-scale datasets that rely heavily on In Situ observations (but may include satellite data), and higher resolution datasets beginning in the 1980s that are based on satellite observations (but may use In Situ data for bias corrections). Satellite-based products are most commonly used for coral bleaching studies due to their higher resolution (e.g., *Liu et al., 2014*), but it is crucial to recognize that important differences exist between these products (e.g., *DeCarlo & Harrison, 2019*), likely resulting from the use of different satellite data and the techniques used for bias correction. Thus, the choice of satellite-based SST product can lead to different conclusions in analyses of the drivers of coral bleaching events (*DeCarlo & Harrison, 2019*).

My analysis of global-mean annual SST anomalies indicates that Coral Reef Watch is a clear outlier from the five other products (Fig. 1). This is evident in both the PCA (Fig. 1C) and pairwise comparisons among products (Figs. 1D–1H; see also Fig. S1). Since the five other products cluster relatively close together (Fig. 1C), even though they are based on partially different source data and different methods, it would be difficult to conclude that the relatively large disparity in Coral Reef Watch represents a more accurate version of SST. Pair-wise comparisons between Coral Reef Watch and the other products show a temporal pattern in which Coral Reef Watch underestimates SST from the mid-1990s to the early 2000s, a pattern that is also represented by the 2nd principal component (Figs. 1D–1H). A similar temporal change in bias between Coral Reef Watch and In Situ loggers was also found on the Great Barrier Reef, in which Coral Reef Watch underestimated reef-water temperatures during 1998, 2002, and 2004 (*DeCarlo & Harrison, 2019*). Additionally, the

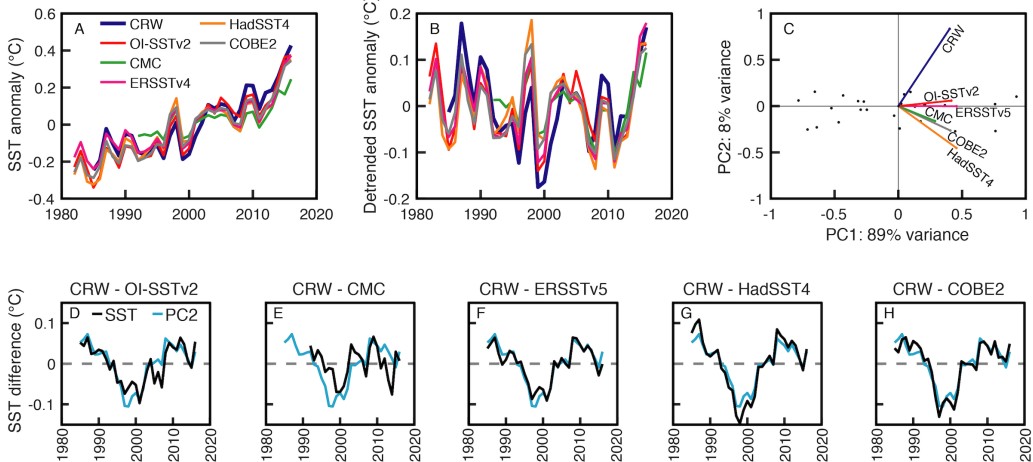

**Figure 1** **Comparison of global-mean annual sea surface temperature (SST) anomalies among various SST products.** (A) Global-mean SST anomalies from 1982–2016, (B) the same data but detrended, and (C) Principal component analysis (PCA) of the detrended annual SST anomalies. (D-H) Pairwise differences of the global-mean anomalies between Coral Reef Watch (CRW) and the other SST products (black) plotted alongside PC2 (light blue). Note that although PC has arbitrary units, it mainly tracks the mismatches between Coral Reef Watch and other products.

root-mean-square error (RMSE) between Coral Reef Watch and other SST products is 0.04–0.07 °C, whereas all other pair-wise combinations have RMSE ≤0.04 °C (Fig. S1). While these differences may seem small, they are sufficient to substantially change DHW, and more detailed analyses on the Great Barrier Reef showed that the differences are exacerbated in the summer months (*DeCarlo & Harrison, 2019*). Thus, even though Coral Reef Watch performs well during the past decade (*DeCarlo et al., 2019*; *DeCarlo & Harrison, 2019*; *Claar, Cobb & Baum, 2019*), I conclude that it has inaccuracies in earlier times that render it inappropriate for multi-decadal analyses of past coral bleaching events. Since the other products are limited by either coarse spatial and temporal resolution (ERSSTv5, HadSST4, COBE2) or temporal coverage (CMC, which only begins in 1992), this leaves OI-SSTv2 as the most suitable SST product to analyze past coral bleaching events globally since 1982.

## The skill of predicting coral bleaching and how to improve it

The numbers of hits, misses, false alarms, and correct negatives vary systematically with the DHW threshold. As this threshold is raised, the numbers of hits decrease, misses increase, false alarms decrease, and correct negatives increase (Fig. 2). This reveals that there are tradeoffs involved in setting the DHW threshold, and these need to be evaluated carefully. For instance, using a low DHW threshold of 2 °C-weeks maximizes the number of hits and minimizes the number of misses, both of which are beneficial features of a forecast. However, such a low DHW threshold produces a large number of false alarms, typically more than half of reef sites each year. These tradeoffs can be quantified in various ways by inspecting how the binary metrics (Table 2) change with the DHW threshold (Figs. 3 and 4). In particular, the Equitable Threat Score (ETS) provides an objective way to define

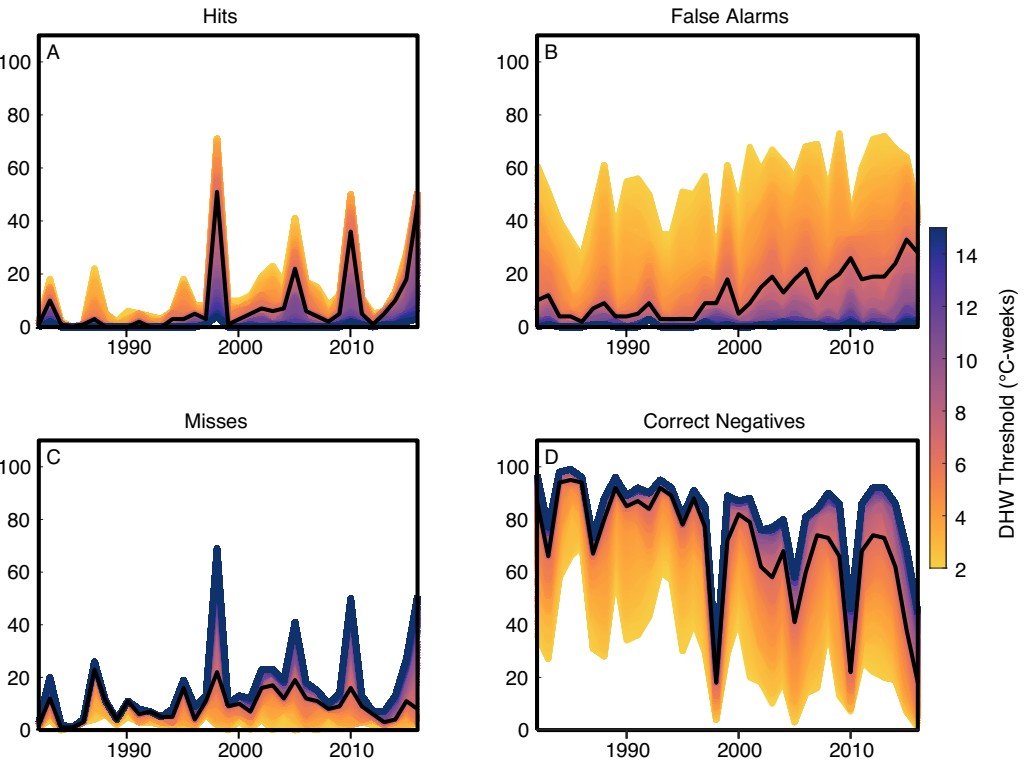

**Figure 2 Reconstructing coral bleaching events with degree heating weeks (DHW).** The four panels show the numbers of (A) hits, (B) false alarms, (C) misses, and (D) correct negatives when bleaching events are predicted for 100 globally-distributed reef systems using various DHW thresholds (colors). The black line indicates the optimal DHW threshold (6.3 °C-weeks) determined by the Equitable Threat Score (ETS; see below).

the optimal DHW, which in this case was 6.3 °C-weeks (when using DHW defined over a 12-week window).

Initial analyses were conducted to determine the best way to define heat stress (Fig. 3 and Table 3). Removing the 1 °C cutoff for accumulating heat stress, and setting the DHW time window to 9 weeks, both improved prediction skill (higher maximum ETS; Fig. 3B). Thus, all further analyses were conducted with DHWs accumulating anytime SST exceeded the MMM over the preceding 9 weeks. With this definition, the optimal DHW for predicting bleaching events was 5.4 °C-weeks. Using either maximum annual Hotspot or MHW decreased prediction skill (Fig. 3B), likely because both metrics represent only magnitude (not duration) of heat stress, and because MHW is not restricted to summer months (i.e., a winter heatwave is unlikely to cause bleaching). Since restricting MHW to summer months would make it effectively the same as Hotspots, and incorporating heatwave duration would make it similar to DHW, there is little utility in the MHW index for coral bleaching. Prediction skill was higher for only severe, rather than all, bleaching events, and the optimal DHW threshold (the DHW at which ETS is maximized) was higher for severe events. These results are consistent with severe bleaching being a result of more intense heat stress, which makes these events easier to separate from non-bleaching events.

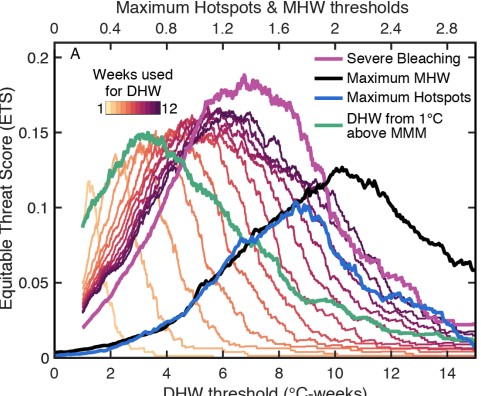
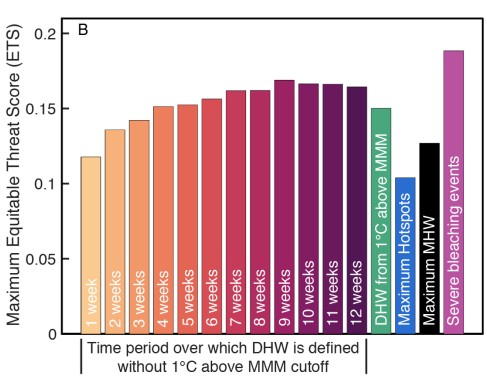

**Figure 3** **ETS for the globally-distributed bleaching events using various predictors and responses.** (A) The analysis was performed with various time windows used to define DHW (yellow to purple colors), DHW including the 1 °C above the MMM cutoff (green; using a 12-week window), maximum Hotspots (light blue), maximum Marine Heatwave (MHW) index (black), or only for severe bleaching events (magenta; using DHW with a 9-week window). Comparison of the green line with the darkest purple line shows the change in absolute value of DHW when including or excluding, respectively, the 1 °C above the MMM cutoff in the DHW definition. (B) Maximum ETS achieved with each of the temperature predictors that were evaluated here.

The ETS varies among regions, with the Indian Ocean and Middle East performing the best, and the Western Atlantic performing the worst (Fig. 4). Evaluations of weather forecasts often consider ETS alongside Bias because ETS tends to reward hits most favorably, and as a result ETS sometimes suggests a threshold with a large positive Bias (i.e., more events are forecast than actually occur). At the optimal DHW threshold, the Bias is relatively close to 1 (0.96) for the Indian Ocean and Middle East, whereas the Bias reaches 1.69 for the Pacific. When all regions are considered together, the ETS is 0.169 and the Bias is 1.27. Using a regional-specific DHW threshold improves the global ETS to 0.2035 (Table 3). As context for these ETS values, weather forecasts predicting rain the following day in the United States and Europe have an ETS of around 0.4, and rain predictions made a week in advance have an ETS of 0.15 (*ECMWF, 2018*). It is important to note that the relatively low ETS of bleaching predictions could reflect a variety of errors. In addition to imperfections in DHW as a predictor, potential errors include inaccuracies in the bleaching database (e.g., unreported bleaching events), or disparities in spatial scales between bleaching and satellite-based SST (i.e., localized warming in reef micro-climates may not be detected in 25-km grid-boxes). Although the potential for bleaching under localized warming is difficult to test retrospectively, if there were more unreported bleaching events earlier in the database, more false alarms would be expected further back in time. However, there are fewer false alarms during 1982–1999 than 2000–2016 (Fig. 2B), even though most reefs were probably monitored less in earlier years. Further, there is no significant trend over time in ETS (Fig. S2) as might be expected if the accuracy of the database improved over time. Therefore, while it remains possible that inaccuracies exist in the database, the patterns in model skill over time do not suggest that this is a major issue.

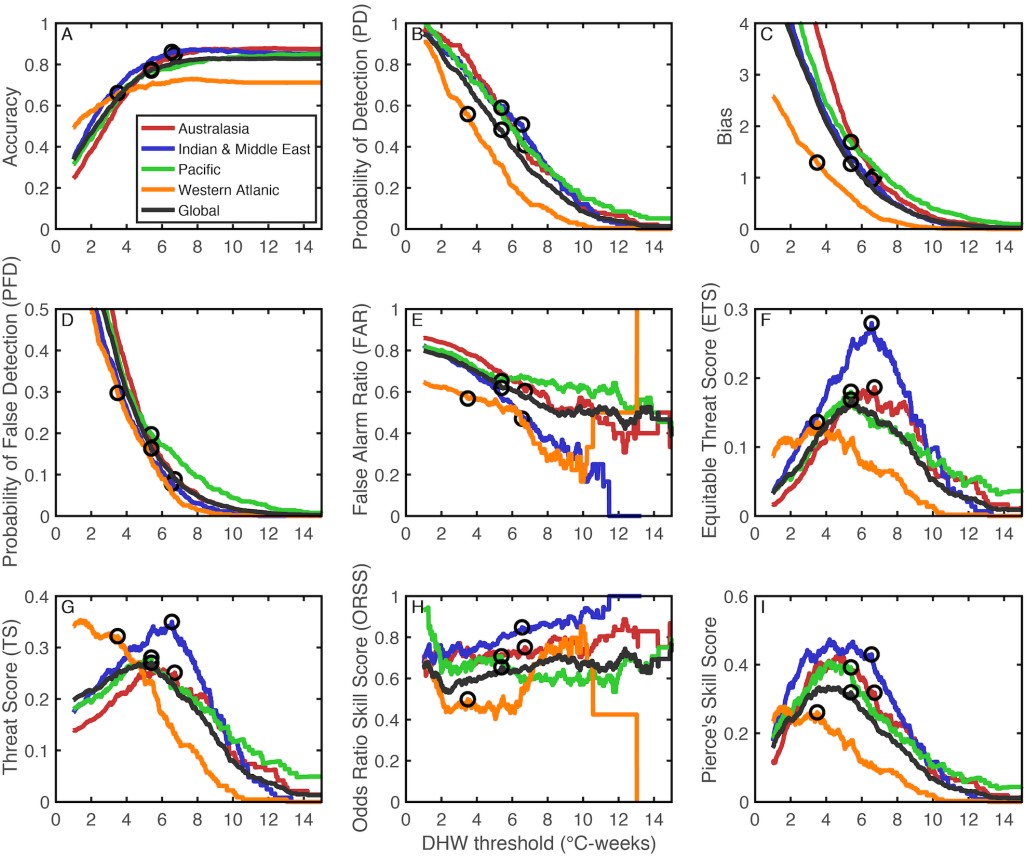

**Figure 4  Forecasting metrics for the skill of reconstructing coral bleaching events from DHW.** The analysis was performed regionally (colored lines) and globally (dark gray line). The definitions of the various metrics (A-I) are listed in Table 2. The black circles indicate the metric values corresponding to the optimal DHW threshold, as determined by the maximum ETS.

One key advantage of the weather forecasting approach is that ETS can be used to objectively test whether other factors modulate the relationship between temperature and bleaching. For example, I tested whether ETS improves with the inclusion of a second predictor: (1) a DHW-threshold trend or (2) El Niño conditions (Fig. 5). A trend in the DHW threshold over time could arise either if corals acclimatize to more frequent heatwaves, or if bleaching events selectively kill the most susceptible species or individuals. For simplicity, I assume any change in the DHW threshold has occurred linearly, although this may be less applicable to changes in heat tolerance following selective mortality events. Previous analyses have concluded that the DHW threshold has increased (*Guest et al., 2012*; *Logan et al., 2014*; *Hughes et al., 2019*; *DeCarlo et al., 2019*; *Sully et al., 2019*). However, the ETS-based analysis presented here indicates that including any trend in the DHW threshold does not improve the ETS (the top 2% of ETS values overlap with zero trend), and therefore the null hypothesis that the DHW threshold has remained constant cannot be rejected. Additionally, El Niño alters the coupled ocean-atmosphere circulation globally, and thus El Niño could affect bleaching independently of temperature through

**Table 3** Changes in ETS and Bias associated with different bleaching predictors and responses.

| Conditions evaluated | Max ETS | ΔETS | Bias at max ETS |
|---|---|---|---|
| Including 1 °C cutoff for DHW[a,b] | 0.1500 | – | 0.9867 |
| Excluding 1 °C cutoff for DHW[a,b,d] | 0.1643 | +0.0143 | 1.1708 |
| DHW defined with 9-week window[a,c,d] | 0.1688 | +0.0045[*] | 1.2670 |
| Including El Niño threshold[a,c,d] | 0.1983 | +0.0295 | 0.8342 |
| Including regional DHW thresholds[a,c,d] | 0.2035 | +0.0338 | 1.2405 |
| Including El Niño and regional DHW thresholds[a,c,d] | 0.2182 | +0.0495 | 0.8242 |
| Only "severe" bleaching events[c,d] | 0.1883 | +0.0195 | 1.4749 |
| Maximum Hotspots[a] | 0.1038 | −0.0650 | 1.4245 |
| Maximum Marine Heatwave index[a] | 0.1268 | −0.0420 | 0.9784 |

Notes.
[a] analysis of bleaching presence (including "moderate" and "severe") versus absence.
[b] DHW defined with 12-week window.
[c] DHW defined with 9-week window.
[d] DHW defined without the 1 °C cutoff.
[*] all subsequent ΔETS are relative to this value.

a variety of mechanisms linked to, for example, winds or clouds (*Smith, 2001*; *DeCarlo et al., 2017*; *Baird et al., 2017*). ETS improves from 0.169 to 0.198 (or to 0.218 if using regional-specific DHW thresholds; Table 3) when including El Niño as an additional threshold, and critically, the highest ETS values are associated with stronger than normal El Niño conditions (Fig. 5). This is clear evidence that El Niño modulates coral bleaching across the tropics through mechanisms beyond just SST because greater skill (higher ETS) is achieved when the El Niño threshold is used in combination with DHW, rather than using DHW alone. Elucidating these non-temperature mechanisms by which El Niño influences bleaching should therefore represent a high priority for future research.

## CONCLUSIONS

Temperature extremes are undoubtedly the primary driver of coral bleaching, but they may not be the only factor involved. Light, nutrients, previous stress history, temperature variance, and rate of warming, among other factors, have all been evoked as contributors to bleaching susceptibility (*Thompson & Van Woesik, 2009*; *Skirving et al., 2017*; *Safaie et al., 2018*; *DeCarlo & Harrison, 2019*; *Lapointe et al., 2019*). While satellite-derived SST data are easy to access, relying on temperature alone to predict bleaching events could lead to overconfidence in its role relative to other factors. Indeed, it may seem surprising that we should put only as much confidence in our ability to reconstruct bleaching events from temperature as we put in rain forecast for next week. The reliability of weather forecasts has improved dramatically over time as a result of continuous evaluation and optimization of the predictor variables. Surely, our ability to predict—and therefore to understand—coral bleaching events can also improve, if critically evaluated within the statistical framework provided by weather forecasting. Removing the 1 °C above the MMM cutoff in the DHW definition, adjusting the DHW window from 12 to 9 weeks, using regional-specific DHW thresholds, and including an El Niño threshold already improves ETS by 45%. My analysis of various predictors is not exhaustive as other heat stress metrics are possible, such as

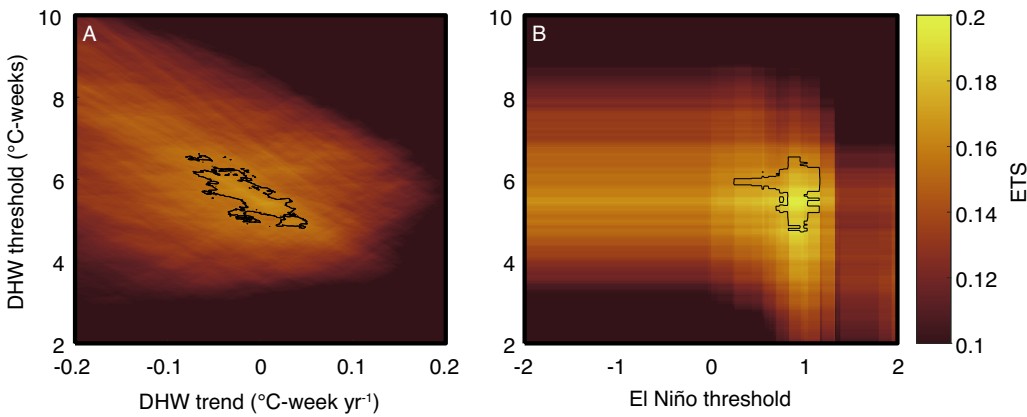

**Figure 5  Evaluation of multivariate predictions of coral bleaching events.** Colors indicate the ETS score for various combinations of the predictors, and black contours show the top 2% of ETS scores. The main analysis was repeated using DHW (with a 9-week window) as a predictor and including a DHW-threshold trend (A), or an El Niño threshold (B). In the trend analysis, the DHW threshold on the y-axis shows the threshold during 1999. The El Niño index is defined as the mean SST anomaly (relative to the 1900–2018 monthly climatology in HadISST; *Rayner et al., 2003*) in the Nino3.4 region, and the threshold is taken as the maximum El Niño index per year.

including temperature variance or the beneficial effects of cooling respites during heat stress (e.g., *Logan et al., 2012*; *Logan et al., 2014*; *Ainsworth et al., 2016*; *Safaie et al., 2018*; *McClanahan et al., 2019*). Rather, the analyses presented here are a demonstration of how ETS can be used to test hypotheses regarding which aspect(s) of temperature and/or other factors are most strongly correlated to bleaching events. Setting our sights on achieving even greater skill will be essential for informing management decisions based on coral sensitivity to heat stress (e.g., *Beyer et al., 2018*; *Darling et al., 2019*).

## Funding
The author received no funding for this work.

## Competing Interests
The author declares that they have no competing interests.

## Author Contributions
- Thomas M. DeCarlo conceived and designed the experiments, analyzed the data, prepared figures and/or tables, authored or reviewed drafts of the paper, and approved the final draft.

## Data Availability
The following information was supplied regarding data availability: All data and code are available at Code Ocean: Thomas DeCarlo (2020) Treating coral bleaching

as weather: a framework to validate and optimize prediction skill [Source Code]. https://doi.org/10.24433/CO.4574023.v2.

## Supplemental Information

Supplemental information for this article can be found online at http://dx.doi.org/10.7717/peerj.9449#supplemental-information.

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
