# Peer review of "Treating coral bleaching as weather: a framework to validate and optimize prediction skill"

_PeerJ, doi:10.7717/peerj.9449_

## Round 0.1 · original submission · Minor Revisions

Dear Dr. DeCarlo,
I hope you can correct those minor revision comments and resubmit you manuscript back. Both reviewers were very positive on this manuscript.
Cheers
Oren Levy

Reviewer 1 ·

Basic reporting

An excellent and highly significant manuscript, well written, appropriately structured with good scholarship and self-contained.

Experimental design

Well described, well presented and rigorous.

Validity of the findings

The conclusions are well stated and reflect the results. My one concern is that weaknesses in the underlying data need to be stated more clearly in the methods and given more prominence in the discussion.

Additional comments

A great piece of research that should inspire much work to improve predictions of bleaching and better understand the factors that drive mass bleaching in addition to temperature.

The Hughes et al 2018 database is the best available but from my understanding of how it was put together it will almost certainly have missed many bleaching events in many of these 100 regions. As you point out, the data base almost certainly miss-classified the first mass bleaching of the modern era which occurred under the noses of an active marine science network on the GBR. You need to mention this and discuss how this might affect your results. You do mention this in brackets at line 206 but it needs more prominence.

Line 101-103. Fisk and Done (1985) is a better reference than Harriott 1985. Her data don’t support her claim of an increase in mortality at Lizard Island and she didn’t see any bleaching. I am sure you are correct in that this was a major bleaching event but only Oliver 1995 addresses the potential scale of the event so this specific criticism of the Hughes et al 2018 database is a little harsh. Perhaps just say that your assessment of these largely anecdotal reports suggest it was a major event. Also, please clarify in the text if this was in 1982 or 1983. My reading suggests that the bleaching occurred in early 1982.

Line 217 – see also Guest et al. (2012)

Line 248+ - when put this way it doesn’t sound like much. Are your values correct ie 7-6 days?

Fisk DA, Done TJ (1985) Taxonomic and bathymetric patterns of bleaching in corals, Myrimidon Reef (Queensland). Proc Fifth Inter Coral Reef Congress 6:149-154
Guest JR, Baird AH, Maynard JA, Muttaqin E, Edwards AJ, Campbell SJ, Yewdall K, Affendi YA, Chou LM (2012) Contrasting patterns of coral bleaching susceptibility in 2010 suggest an adaptive response to thermal stress. Plos One 7:e33353

Reviewer 2 ·

Basic reporting

The manuscript by DeCarlo outlines an elegant improvement of predictions of coral bleaching. This is the type of work NOAA should be doing, but full credit goes to DeCarlo for taking the lead and fixing the problem. This manuscript should be definitely published with some minor changes. The manuscript was very well written. The literature was contemporary and appropriate. The code for the figures and the calculations were shared and accessible. Hypotheses were relevant, although I would have liked a few more sentences on methods tried that didn't improve the model skill.

Line 22. Change the main results in the Abstract to be more assertive and specific.

Change: “Adjustments to the degree heating weeks (DHW) metric, regional DHW
thresholds, and adding an El Niño threshold, together resulted in a 45% increase in prediction skill relative to standard methods.”

to

“Removing the 1 °C above the maximum monthly mean cutoff in the degree heating weeks (DHW) definition, adjusting the DHW window from 12 to 9 weeks, using regional-specific DHW thresholds, and including an El Niño threshold already improves the model skill by 45%.

Line 110 to 114. I’d like to see equations regarding the different tests that were run to assess skill (I guess Table 2 works, but some equations in the text would clarify the methods).
Line 116. Could use a table here with some details.
Line 161, should read “is”, not “in”
Lines 168 to 177. This text should be in the methods. These are not results.
Line 182. This sentence should be in the Abstract.
Line 206. Include a comment that bleaching temperatures may change over time here.
Figure 1 was overloaded. It was difficult to follow. Best to place all of the pairwise figures in the supplementary document.

Figure 3. It was not immediately clear what to focus on in Figure 3. It took me far too long to get the point of this figure. Best to try a different approach. The DHW colors seemed to work but things get messy with the gray, black and dashed lines. Pretend you are giving a seminar and that you have to get your point across within less than 20 seconds. What do you want the viewer (i.e., reader in this context) to focus on? Highlight that point in the figure.

Experimental design

The manuscript outlines an important piece of original research. The rationale was clear and the work fills knowledge gaps. The calculations were appropriate for the research. The analysis was a sophisticated approach to an important problem. The details of the methods was appropriate.

Validity of the findings

The results are extremely useful and should be applied globally. Figures 1 and 3 need some refinement.

Additional comments

The DeCarlo manuscript was well written and is an excellent contribution to the literature.

---

## Round 0.2 · accepted · Accept

Congratulations, well done!